# Bayesian Inference for Large Scale Image Classification

## Abstract

Bayesian inference promises to ground and improve the performance of deep neural networks. It promises to be robust to overfitting, to simplify the training procedure and the space of hyperparameters, and to provide a calibrated measure of uncertainty that can enhance decision making, agent exploration and prediction fairness. Markov Chain Monte Carlo (MCMC) methods enable Bayesian inference by generating samples from the posterior distribution over model parameters. Despite the theoretical advantages of Bayesian inference and the similarity between MCMC and optimization methods, the performance of sampling methods has so far lagged behind optimization methods for large scale deep learning tasks. We aim to fill this gap and introduce ATMC, an adaptive noise MCMC algorithm that estimates and is able to sample from the posterior of a neural network. ATMC dynamically adjusts the amount of momentum and noise applied to each parameter update in order to compensate for the use of stochastic gradients. We use a ResNet architecture without batch normalization to test ATMC on the Cifar10 benchmark and the large scale ImageNet benchmark and show that, despite the absence of batch normalization, ATMC outperforms a strong optimization baseline in terms of both classification accuracy and test log-likelihood. We show that ATMC is intrinsically robust to overfitting on the training data and that ATMC provides a better calibrated measure of uncertainty compared to the optimization baseline.

## 1 Introduction

In contrast to optimization approaches in machine learning that derive a single estimate for the weights of a neural network, Bayesian inference aims at deriving a posterior distribution over the weights of the network. This makes it possible to sample model instances from the distribution over the weights and offers unique advantages. Multiple model instances can be aggregated to obtain robust uncertainty estimates over the network's predictions; uncertainty estimates are crucial in domains such as medical diagnosis and autonomous driving where following a model's incorrect predictions can result in catastrophe (Kendall & Gal, 2017). Sampling a distribution, as opposed to optimizing a loss, is less prone to overfitting and more training doesn't decrease test performance. Bayesian inference can also be applied to differential privacy, where each individual sample has increased privacy guarantees (Wang et al., 2015), and to reinforcement learning, where one can leverage model uncertainty to balance between exploration and exploitation (Osband & Van Roy, 2017).

Traditional Markov Chain Monte Carlo (MCMC) methods like HMC (Neal et al., 2011) are a standard class of methods for generating samples from the posterior distribution over model parameters. These methods are seldom applied in deep learning because they have traditionally failed to scale well with large datasets and many parameters (Rajaratnam & Sparks, 2015). Stochastic Gradient MCMC (SG-MCMC) methods have fared somewhat better in scaling to large datasets due to their close relationship to stochastic optimization methods. For example the SGLD sampler (Welling & Teh, 2011) amounts to performing stochastic gradient descent while adding Gaussian noise to each parameter update. Despite these improvements, samplers like SGLD are only guaranteed to converge to the correct

---

**Algorithm 1** The ATMC sampler. The algorithm accepts the initialized model parameters $\boldsymbol{\theta}_0$, step size $h$, pre-conditioner $m$, and momentum noise $D_c$.

---

1: **procedure** ATMC_TRAINING($\boldsymbol{\theta}_0, h, m, D$)
2:     $\boldsymbol{p}_0 \leftarrow 0$
3:     $\boldsymbol{\xi}_0 \leftarrow 0$
4:     **while** $t < T$ **do**
5:         $\boldsymbol{G}_t \leftarrow minibatch\_gradient(\boldsymbol{\theta}_t)$
6:         $\boldsymbol{\eta}_t \leftarrow random\_normal()$
7:         $\boldsymbol{\alpha}_t \leftarrow \max(D - \boldsymbol{\xi}_t, 0)$
8:         $\boldsymbol{\beta}_t \leftarrow \boldsymbol{\alpha}_t + \boldsymbol{\xi}_t$
9:         $\boldsymbol{p}_{t+h} \leftarrow e^{\boldsymbol{\beta}_t h}\left[\boldsymbol{p}_t - \frac{\exp[\boldsymbol{\beta}_t h] - 1}{\boldsymbol{\beta}_t}\boldsymbol{G}_t + \sqrt{\frac{\exp[2\boldsymbol{\beta}_t h] - 1}{\boldsymbol{\beta}_t}\boldsymbol{\alpha}_t}\,\boldsymbol{\eta}_t\right]$
10:       $\boldsymbol{\theta}_{t+h} \leftarrow \boldsymbol{\theta}_t + h\frac{\boldsymbol{p}_{t+h}}{m}$
11:       $\boldsymbol{\xi}_{t+h} \leftarrow \boldsymbol{\xi}_t + h\left[\frac{\boldsymbol{p}_{t+h}^2}{m} - 1\right]$
12:       $t \leftarrow t + h$

---

distribution when the step size is annealed to zero; additional control variates have been developed to mitigate this to some extent (Ahn et al., 2012; Ding et al., 2014).

The objective of this work is to make Bayesian inference practical for deep learning by making SG-MCMC methods scale to large models and datasets. The contributions described in this work fall in three categories. We first propose the Adaptive Thermostat Monte Carlo (ATMC) sampler that offers improved convergence and stability. ATMC dynamically adjusts the amount of momentum and noise applied to each model parameter. Secondly, we improve an existing second order numerical integration method that is needed for the ATMC sampler. Third, since ATMC, like other SG-MCMC samplers, is not directly compatible with stochastic regularization methods such as batch normalization (BatchNorm) and Dropout (see Sect. 4), we construct the ResNet++ network by taking the original ResNet architecture (He et al., 2016), removing BatchNorm and introducing SELUs (Klambauer et al., 2017), Fixup initialization (Zhang et al., 2019a) and weight normalization (Salimans & Kingma, 2016). We design ResNet++ so that its parameters are easy to sample from and the gradients are well-behaved even in the absence of BatchNorm.

We show that the ATMC sampler is able to outperform optimization methods in terms of accuracy, log-likelihood and uncertainty calibration in the following settings. First, when using the ResNet++ architecture for both the ATMC sampler and the optimization baseline, the ATMC sampler significantly outperforms the optimization baseline on both Cifar-10 and ImageNet. Secondly, when using the standard ResNet for the optimization baseline and the ResNet++ for the ATMC sampler, multiple samples of the ATMC that approximate the predictive posterior of the model are still able to outperform the optimization baseline on ImageNet. Using the ResNet++ architecture, the ATMC sampler reduces the need for hyper-parameter tuning since it does not require early stopping, does not use stochastic regularization, is not prone to over-fitting on the training data and avoids a carefully tuned learning rate decay schedule.

## 2 ATMC SAMPLER

In this section we define the Stochastic Differential Equation (SDE) that gives rise to the ATMC sampler described in Algorithm 1. A detailed background and framework for constructing SDEs that converge to a target distribution can be found in (Ma et al., 2015).

### 2.1 GENERAL FORM OF THE SDE

Our starting point for constructing the ATMC sampler is the framework of Stochastic Differential Equations. We are interested in SDEs that converge to a distribution $p(z)$ over the vector $z \in \mathbb{R}^d$ for which we can evaluate $\nabla \log p(z)$. Because only the gradient of $\log p(z)$

is required, it is sufficient to define an energy function $H(z) = -\log p(z) + C$ up to a constant $C$. As a consequence, we can sample from the posterior distribution $p(\theta|x)$ by only evaluating the energy function gradient $\nabla H(\theta) = -\nabla \log p(x, \theta)$. The general form of SDEs converging to $p(z)$ for which only the gradient of $p(z)$ is required is as follows (Ma et al., 2015):

$$dz = -\left[D(z) + Q(z)\right]\nabla H(z)dt + \Gamma(z)dt + \sqrt{2D(z)}dW_t, \quad \Gamma_i(z) = \sum_j^d \frac{\partial\left[D_{ij}(z) + Q_{ij}(z)\right]}{\partial z_j},$$
(1)

where $D(z)$ is a positive-definite matrix that determines the amount of noise, $Q(z)$ is a skew-symmetric matrix that mixes energy between variables, $W_t$ is a Wiener process, and $\Gamma(z)$ is a correction factor that compensates for dynamics that depend on the current state $z$. The ATMC sampler that we propose is an instance of (1) for specific definitions of $H(z)$, $D(z)$, and $Q(z)$.

## 2.2 Energy Function

We start by defining the energy function $H(z)$. The energy function for the model posterior $p(\theta|x)$ is defined by the loss function $\mathcal{L}(\theta) = -\log p(x, \theta)$. Because the dataset $x$ is generally large, we would like to only evaluate a mini-batch loss $\tilde{\mathcal{L}}(\theta)$. However, naively using a stochastic gradient in (1) will result in significant bias (Chen et al., 2014). Motivated by the Central Limit Theorem, the stochastic gradient is assumed to follow a Gaussian distribution $\nabla\tilde{\mathcal{L}}(\theta) \sim \mathcal{N}(\nabla\mathcal{L}(\theta), B)$ where the covariance $B$ is additionally assumed to be diagonal and constant w.r.t. $\theta$. The energy function for the ATMC sampler is defined as:

$$H(\theta, p, \xi) = \mathcal{L}(\theta) + K(p) + \frac{1}{2}\left(\xi - \frac{\text{diag}(B)}{2m}\right)^2,$$
(2)

where $p$ is the momentum, $K(p)$ defines the momentum distribution, and $\xi$ is a control variate referred to as the temperature. Both $p$ and $\xi$ have the same dimensionality as $\theta$. The hyper-parameter $m$ controls the strength of the coupling between $\xi$ and $p$. The distribution of the control variate $p(\xi)$ depends on the amount of noise $B$ in the stochastic gradient estimate $\tilde{\mathcal{L}}(\theta)$.

## 2.3 Noise robust dynamics

Next we define the dynamics $Q(z)$ and $D(z)$ such that the SDE that results from (1) can be simulated without the need to evaluate $B$:

$$D(\theta, p, \xi) = \begin{pmatrix} 0 & 0 & 0 \\ 0 & \alpha(\xi)m + \frac{1}{2}B & 0 \\ 0 & 0 & 0 \end{pmatrix}, \quad Q(\theta, p, \xi) = \begin{pmatrix} 0 & -I & 0 \\ I & 0 & m\nabla K(p) \\ 0 & -m\nabla K(p) & 0 \end{pmatrix}, \quad (3)$$

where $\alpha(\xi)$ is a non-negative function that determines how the temperature $\xi$ affects the amount of noise added to the momentum update.

We first illustrate the resulting SDE by using a simpler Gaussian momentum distribution $K(p) = \|p\|^2/(2m)$. Note that the variance of the momentum $\text{Var}(p) = m$ is reused in (2) and (3) to control the strength of the coupling between $\xi$ and $p$. This will result in a temperature control with a momentum friction term proportional to $\xi$, unlike previously reported thermostat MCMC methods (Ding et al., 2014; Lu et al., 2016) where the friction

term is proportional to $\xi/m$. We substitute the dynamics $Q(z)$ and $D(z)$ defined in (3) and energy function $H(z)$ defined in (2) into (1):

$$\begin{pmatrix} d\theta \\ dp \\ d\xi \end{pmatrix} = \begin{pmatrix} p/m \\ -\nabla\tilde{\mathcal{L}}(\theta) - \beta(\xi)p \\ p^2/m - 1 \end{pmatrix} dt + \begin{pmatrix} 0 & 0 & 0 \\ 0 & \sqrt{2\alpha(\xi)m} & 0 \\ 0 & 0 & 0 \end{pmatrix} dW_t, \quad \beta(\xi) = \alpha(\xi) + \xi, \quad (4)$$

where we use $\nabla\tilde{\mathcal{L}}(\theta)dt = \nabla\mathcal{L}(\theta)dt + \sqrt{B}dW_t$ to replace the gradient of the loss with the mini-batch estimate. The momentum $p$ is dampened by a friction term $\beta(\xi)$ that depends on the choice of $\alpha(\xi)$. The stochastic gradient noise $B$ does not show up in (4) due to the particular choice of energy function $H(z)$ and dynamics $Q(z), D(z)$. Note however this analysis relies on the assumption that the covariance of the stochastic gradient noise $B$ is constant in $\theta$ and a single temperature variable per parameter can only correct for a diagonal covariance $B$. We do not expect that this assumption will hold in practice and the approximation will therefore lead to bias in the samples. However, annealing the step size $h$ will reduce the error due to mini-batching together with other sources of discretization error (Welling & Teh, 2011).

## 2.4 Adaptive Noise Thermostat

Finally, we must choose a function $\alpha(\xi)$ which controls the amount of noise and momentum damping $\beta(\xi)$. Previous work uses the Nosé-Hoover thermostat that is defined by $\alpha(\xi) = D_c$ where $D_c$ is a constant determining the amount of noise added to the momentum update (Ding et al., 2014). Although the Nosé-Hoover thermostat is able to correct the stochastic gradient noise $B$, the correction comes at the cost of slower convergence because additional friction $\beta(\xi)$ is applied as $B$ increases. Another drawback of the Nosé-Hoover thermostat is that it causes negative friction when $\xi < -D_c$. In the negative friction phase $\beta(\xi) < 0$, previous gradient terms are amplified rather than dampened. Although this behavior is mathematically sound we find that it can cause exploding momentum variables.

Our choice of $\alpha(\xi)$ is based on the idea that negative friction should not occur and convergence speed should not be reduced by the stochastic gradient noise. Based on this intuition, we define the ATMC sampler by $\alpha(\xi) = \max(D - \xi, 0)$. The ATMC sampler is best characterized by the various temperature stages. For $0 < \xi < D_c$ the total amount of noise added to the momentum is $D_c$ and the friction coefficient $\beta(\xi) = D_c$. At this stage, the stochastic gradient noise is compensated for by adding less noise to the momentum update. If $B \gg D_c$ the dominant stage will be $\xi > D$ resulting in $\beta(\xi) < D_c$ and zero noise being added to the momentum. Finally, when $\xi < 0$ the friction coefficient $\beta(\xi) = D_c$ and the noise added to the momentum is proportional to $D_c - \xi$. Thus, the momentum always experiences a minimum amount of friction $\beta(\xi) \geq D_c$ determined by the hyper-parameter $D_c$ and the noise added to the momentum update is automatically adjusted based on the amount of noise present in the stochastic gradients.

## 2.5 Momentum energy function

Following (Lu et al., 2016), we generalize the momentum energy function $K(p)$ to the symmetric hyperbolic distribution which is defined as follows (Lu et al., 2016):

$$K(p) = \sum_i mc^2 \left[ \sqrt{\frac{p_i^2}{m^2c^2} + 1} - 1 \right], \quad (5)$$

where $m$ and $c$ are hyper-parameters. The Gaussian kinetic energy $K(p) = \|p\|^2/(2m)$ is a special case obtained by taking the limit $c \to \infty$. The magnitude of parameter updates $\|\Delta\theta\|$ is determined by the gradient of the momentum:

$$\|\Delta\theta\| = \|\nabla K(p)\| = \left\| \frac{p}{M(p)} \right\|, \quad M(p) = m\sqrt{\frac{p^2}{m^2c^2} + 1}. \quad (6)$$

Hence, the hyperbolic distribution results in relativistic momentum dynamics where the parameter updates are upper bounded by $c$ and the pre-conditioner $M(p)$ depends on $p$. The average update magnitude $\mathcal{E}[\|\nabla K(p)\|] \approx 1/\sqrt{m}$ for $c \gg m$. Consequently, the parameters $m$ and $c$ are interpretable hyper-parameters controlling the average and maximum parameter update per step together with the step size $h$.

The SDE we derive in (4) and integrate in Sec. 3 uses a Gaussian momentum energy function for clarity. Deriving ATMC with a different momentum distribution like the hyperbolic distribution amounts to substituting (2), (3), and the alternative momentum distribution into (1). For the hyperbolic distribution, the dynamic friction coefficient $\beta(\xi)$ will also depend on $p$. For the numerical integration of (4) with a hyperbolic momentum distribution we assume $\beta(\xi)$ to be constant in $p$.

## 3 Improved numerical integrator for MCMC samplers

In this section we construct the numerical integrator required to numerically approximate the ATMC sampler defined in (4). An efficient numerical integrator can be constructed by splitting the SDE into two terms:

$$
\begin{pmatrix} d\theta \\ dp \\ d\xi \end{pmatrix} = \underbrace{\begin{pmatrix} p/m \\ 0 \\ p^2/m - 1 \end{pmatrix} dt}_{A} + \underbrace{\begin{pmatrix} 0 \\ -\nabla\tilde{\mathcal{L}}(\theta) - \beta(p,\xi)p \\ 0 \end{pmatrix} dt + \begin{pmatrix} 0 & 0 & 0 \\ 0 & \sqrt{2\alpha(\xi)m} & 0 \\ 0 & 0 & 0 \end{pmatrix} dW_t}_{B}. \quad (7)
$$

Hence, we obtain a linear ODE in part (A) that updates the parameters $\theta$ and the thermostats $\xi$ and a linear SDE in part (B) that updates the momentum $p$. The operators that simulate these dynamics exactly for a time step $h$ are denoted $\phi_A^h$ and $\phi_B^h$, respectively. Using the Strang splitting scheme yields a second order method (Chen et al., 2015):

$$
\phi^h = \phi_B^{h/2} \circ \phi_A^h \circ \phi_B^{h/2}. \quad (8)
$$

The first operator $\phi_A^h$ is given by

$$
\phi_A^h(z_t) = \left( \theta_t + h\frac{p_t}{m} \quad p_t \quad \xi_t + h\left[\frac{p_t^2}{m} - 1\right] \right)^T. \quad (9)
$$

The second operator $\phi_B$ is an instance of the Ornstein–Uhlenbeck process which can also be computed analytically as follows (Nelson, 1967):

$$
\phi_B^h(z_t) = \left( \theta_t \quad e^{\beta(\xi_t)h}\left[ p_t - \gamma_1(\xi_t)\nabla\tilde{\mathcal{L}}(\theta_t) + \sqrt{\gamma_2(\xi_t)\alpha(\xi_t)}\,\eta_t \right] \quad \xi_t \right)^T,
$$
$$
\gamma_a(\xi_t) = \frac{\exp[a\,\beta(\xi_t)h] - 1}{\beta(\xi_t)}, \quad (10)
$$

where $\eta_t$ is isotropic Gaussian noise. Previous work (Chen et al., 2015) on higher order integrators for samplers splits the SDE into three parts where the third term is obtained from separating the friction term from the other terms in the momentum update $\phi_B$. By integrating (10) exactly the gradient step and the noise and gradient term are directly affected by the friction. An exact momentum update provides additional robustness to large gradients because the temperature will increase in order to compensate for momentum updates that would lead to excessively large steps. Another advantage of a two-way split integrator is that the first and last steps in (8) can be fused together such that only a momentum update is performed per iteration. Algorithm 1 shows the pseudocode for the ATMC sampler with the split integrator defined in (9) and (10).

## 4 The ResNet++ Architecture

The generalization performance of large neural nets trained using optimization depend on stochastic regularization methods like Dropout (Srivastava et al., 2014) and BatchNorm

| $x$ | Conv 1x1 | BatchNorm | ReLU | Conv 3x3 | BatchNorm | ReLU | Conv 1x1 | BatchNorm | $+x$ | ReLU |
|---|---|---|---|---|---|---|---|---|---|---|

| $x$ | Conv 1x1 | SeLU | Conv 3x3 | SeLU | Conv 1x1 | $+x$ | SeLU |
|---|---|---|---|---|---|---|---|

Figure 1: Residual blocks in respectively the ResNet and ResNet++ architectures.

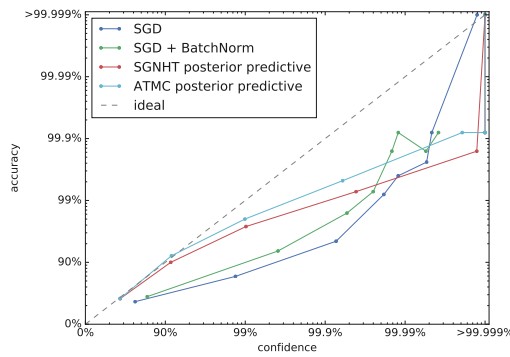

Figure 2: Calibration plot for Cifar10

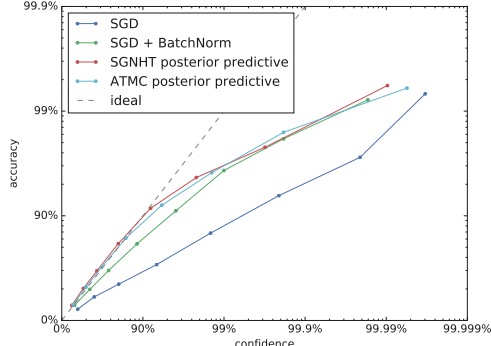

Figure 3: Calibration plot for ImageNet

(Ioffe & Szegedy, 2015). These methods implicitly add noise into the model parameters (Kingma et al., 2015; Teye et al., 2018) and significantly boost training performance and generalization for image classifiers. These methods can be interpreted as a coarse approximation of Bayesian Inference (Kingma et al., 2015; Teye et al., 2018). But a stochastic gradient sampler like ATMC already adds the necessary amount of noise and combined with BatchNorm or Dropout it leads to underfitting. We thus define a BatchNorm free version of ResNet called ResNet++ that includes SELUs (Klambauer et al., 2017), Fixup initialization (Zhang et al., 2019a) and weight normalization (Salimans & Kingma, 2016) (see Fig. 1). We use ATMC to fill the significant gap in performance due to the absence of BatchNorm in ResNet++.

## 4.1 SELU

We find the SELU activation to work well in BatchNorm free networks. SELU forces the statistics of the activations towards zero mean and unit variance (Klambauer et al., 2017). The SELU activation function additionally has a non-zero gradient everywhere which could improve the mixing of the sampler by providing a more informative gradient.

## 4.2 Fixup initialization

ResNets are known to scale well with depth (He et al., 2016). However, the additive effect of the residual branch causes the magnitudes of the activations to increase with the number of residual connections. Fixup is a recently proposed initialization method that mitigates the exploding residual branch problem without using BatchNorm (Zhang et al., 2019a). We use a simplified version of Fixup by initializing the scales of the final layer in each residual branch to a small constant.

## 4.3 Weight normalization

We use weight normalization (Salimans & Kingma, 2016) to separate the direction and scale of each linear feature vector

$$\theta^{(i)} = \phi_s^{(i)} \frac{\phi_d^{(i)}}{\left\| \phi_d^{(i)} \right\|},$$

(11)

Table 1: Performance on Cifar10 with ResNet-56 model. The posterior predictive is estimate using a sample of the posterior parameters at the end of each learning rate cycle.

| Setup | Top 1 acc. [%] | NLL [Nats] |
|---|---|---|
| SGD | 91.5 | 0.370 |
| SGD + BatchNorm | **94.4** | 0.243 |
| ATMC (single sample) | 92.4 | 0.303 |
| ATMC (Posterior predictive) | 93.9 | **0.194** |
| SGNHT (single sample) | 91.7 | 0.343 |
| SGNHT (Posterior predictive) | 93.5 | 0.211 |

where $\phi_d^{(i)}$ is the direction vector and $\phi_s^{(i)}$ is the magnitude of a feature vector $\theta^{(i)}$. Weight normalization does not depend on batch statistics and is compatible with MCMC methods.

The scale of the direction vector does not affect the outputs of the model. It does however affect the effective step size (Wu et al., 2018). Therefore the prior on the direction vector $\phi_d^{(i)}$ is chosen such that it is forced to unit length

$$p(\phi_d^{(i)}) \propto \exp\left[-\frac{d}{2}\left(\left\|\phi_d^{(i)}\right\|^2 - 1\right)^2\right].$$ (12)

The prior on the scales $p(\phi_s)$ is problem-specific and can for example be chosen to encode a preference for structurally sparse models.

## 5 EXPERIMENTS

The experiments presented here aim to demonstrate that the ATMC sampler is competitive with a well-tuned optimization baseline for large-scale datasets and models. We use the TensorFlow official implementation of ResNet-56 and ResNet-50 on Cifar10 and ImageNet, respectively. We compare our ATMC sampler to an optimization baseline with and without BatchNorm. For the optimization baseline without BatchNorm we use the ResNet++ architecture as described in Sec. 4. For the baseline with BatchNorm we found standard ResNet with Xavier initialization and the ReLU non-linearity to work better.

For the ATMC sampler we report both the performance of a single sample and the estimated posterior predictive based on a finite number of samples. Similar to earlier work (Zhang et al., 2019b) we found that many fewer samples are needed when a cyclic step size $h_t = h_0 * \frac{1}{2}[1 + \cos(\pi \mod[t, n])]$ with cycle length $n$ is used. The final sample in each cycle is used to estimate the posterior predictive.

For ResNet++ we further use a group Laplace prior $p(\theta_i) \propto \exp(-\|\theta_i\|/b)$ with $b = 5$ to regularize the scales of each linear feature in ResNet++. The momentum noise is chosen as $D_c = -\log(0.9)/h_0$ such that the friction applied to the momentum is at least 0.9.

### 5.1 CIFAR 10

For Cifar10 we choose the step size $h_0 = 0.001$ and the cycle length is set to 50 epochs. The momentum hyper-parameters are $m = (0.0003/h_0)^{-2}$ and $c = 0.001/h_0$ such that the average speed and maximum speed per step are 0.0003 and 0.001, respectively. The number of convolution filters is doubled to 32 compared to the original ResNet-56 implementation. We use a single V100 GPU with a batch size of 128. The sampler runs for 1000 epochs and we start collecting samples for the posterior predictive after 150 epochs. The optimization baseline converges in 180 epochs. We also report the results of sampling with a sampler based Nosé-Hoover thermostats (SGNHT) (Ding et al., 2014; Lu et al., 2016) applied to the ResNet++ architecture.

Table 2: Performance on ImageNet with ResNet-50 model. The posterior predictive is estimated using a sample of the posterior parameters at the end of each learning rate cycle.

| Setup | Top 1 acc. [%] | NLL [Nats] |
|---|---|---|
| SGD | 70.9 | 1.24 |
| SGD + BatchNorm | 76.2 | 0.947 |
| ATMC (single sample) | 74.2 | 1.08 |
| ATMC (Posterior predictive) | **77.5** | **0.883** |
| SGNHT (single sample) | 73.1 | 1.15 |
| SGNHT (Posterior predictive) | 76.4 | 0.941 |

Table 1 lists the test set performance for Cifar10. A single sample from the posterior already outperforms the baseline without BatchNorm by a significant margin in both test accuracy and log-likelihood. Using BatchNorm significantly improves the generalization of the optimization baseline. It outperforms the estimate of the posterior predictive in accuracy yet it does not have a better test log-likelihood.

To further analyze the quality of the uncertainty estimates, we group each model's prediction in 8 equally sized bins based on the confidence $p(\hat{\omega}_i|x_i)$ where $\hat{\omega}_i$ is the maximum probability class for example $x_i$. If the probabilities are well-calibrated, the average confidence should be close to the average accuracy. Figure 2 shows the calibration of the uncertainty estimates for the posterior predictive and optimization baselines. The posterior predictive is calibrated for the least confident predictions $p(\hat{\omega}_i|x_i) < 0.9$ and shows less bias towards overconfidence compared to the models trained with SGD.

## 5.2 ImageNet

For the ImageNet experiments we use an initial step size $h_0 = 0.0005$ and a cycle length of 20 epochs. The other hyper-parameters for the sampler are the same as for the Cifar10 experiments. We use a a single Google Cloud TPUv3 with a batch size of 1024. We did not observe a significant difference in wall clock time per training step between SGD and ATMC on the same model. Each training step using ResNet+BatchNorm model takes 20% longer in wall clock time compared to a single train step using ResNet++. Samples for the posterior predictive are collected after 150 epochs and the sampler runs for 1000 epochs. The optimization baseline converges in 90 epochs.

Table 2 lists the results for ImageNet classification. A single sample from the posterior outperforms the optimization baseline without BatchNorm. The posterior predictive based on ATMC outperforms the optimizer with BatchNorm by a wide margin in both accuracy and test log-likelihood. We note that the sampler runs significantly longer (10x) compared to the optimization baseline because it takes a long time for the posterior predictive estimate to converge. However, the posterior predictive of ATMC matches the accuracy of the optimization baseline with BatchNorm (76.2%) after 240 epochs.

Figure 3 shows the quality of the uncertainty for various levels of confidence. Again, the ATMC based posterior predictive produces much better calibrated predictions and is almost perfectly calibrated for low confidence predictions $p(\hat{\omega}_i|x_i) < 0.9$ and shows less bias towards overconfidence compared to the optimization baseline.

## 6 Discussion

The empirical results show it is possible to sample the posterior distribution of neural networks on large scale image classification problems like ImageNet. A major obstacle for sampling the posterior of ResNets in particular is the lack of compatibility with Batch-Norm. Using recent advances in initialization and the SELU activation function we are able to stabilize and speed up training of ResNets without resorting to BatchNorm. Nonetheless, we observe that BatchNorm still offers a unique advantage in terms of generalization per-

formance. We hope that future work will allow the implicit inductive bias that BatchNorm has to be transferred into an explicit prior that is compatible with sampling methods.

Multiple posterior samples provide a much more accurate estimate of the posterior predictive, and consequently much better accuracy and uncertainty estimates. For inference, making predictions using a large ensemble of models sampled from the posterior can be costly. Variational Inference methods can be used to quickly characterize a local mode of the posterior (Blundell et al., 2015). More recent work shows that a running estimate of the mean and variance of the parameters during training can also be used to approximate a mode of the posterior (Maddox et al., 2019). Methods like distillation could potentially be used to compress a high-quality ensemble into a single network with a limited computational budget (Balan et al., 2015).

Although the form in (4) is very general, alternative methods for dealing with stochastic gradients have been proposed in the literature. One approach is to estimate the covariance of the stochastic gradient noise $B$ explicitly and use it correct and pre-condition the sampling dynamics (Ahn et al., 2012; Li et al., 2016).

Other sampling methods are not based on an SDE that converges to the target distribution. Under some conditions stochastic optimization methods can be interpreted as such a biased sampling method (Mandt et al., 2017). Predictions based on multiple samples from the trajectory of SGD have been used successfully for obtaining uncertainty estimates in large scale Deep Learning (Maddox et al., 2019). However, these methods rely on tuning hyperparameters in such a way that just the right amount of noise is inserted.

## 7    Conclusion

This work introduces the ATMC sampler, a robust posterior sampling method that scales to large deep learning problems. To the best of our knowledge, we are the first to successfully train neural networks using MCMC on ImageNet. In a BatchNorm free setting, a single sample from the posterior generated by ATMC outperforms the optimization baseline. A posterior predictive estimate outperforms the optimization baseline with BatchNorm on ImageNet. Based on these empirical results we hope the ATMC sampler will enable new applications of Bayesian inference in deep learning.

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
