# OpenReview forum: "Bayesian Inference for Large Scale Image Classification"
_ICLR.cc/2020/Conference — Reject_

### Official Review · AnonReviewer2 · 2019-10-22
**Official Blind Review #2**

**Rating:** 6

**Review:**

Summary:

The main idea of the paper is the introduction of ATMC, an adaptive noise MCMC algorithm that dynamically adjusts the momentum and noise applied to each parameter update while sampling from the posterior of a neural network. A modified version of ResNet architecture called ResNet++ has been introduced in the paper and later used to train it on ImageNet using MCMC which is great work considering the complexities associated. Furthermore, the authors claim ATMC to be robust to overfitting and it provides a measure of uncertainty with predictions. The paper is well written with clear flow and good mathematical explanation.

Questions from authors:

1. The approach of Stochastic Gradient MCMC (SG-MCMC) has been there for quite some time and in this paper, there is no clear explanation of the advantages of ATMC over SG-MCMC methods. The authors can argue that the introduction of dynamically adjustable momentum and noise can make the paper novel and unique. However, in Stochastic gradient Markov chain Monte Carlo (SG-MCMC) [1], the authors used a meta-learning algorithm to learn Hamiltonian dynamics with state-dependent drift and diffusion which makes the system scalable to large datasets, very similar to this work. Also, changing the momentum variable in Hamiltonian dynamics with thermostat variable in stochastic gradient thermostats, the meta learner algorithm of [1] can be used to make the stochastic gradient thermostats approach adaptive. What are the advantages of the approach proposed in this paper over [1]?

2.  The paper follows the approach of stochastic gradient thermostats as introduced in the paper  [2] and builds upon it to make it adaptive to improve stability and for better convergence.  The advantages are clearly mentioned in the paper except for the case when 0 < ξ < D_c the total amount of noise added to the momentum is D_c and the friction coefficient β(ξ) = D_c. At this stage, the authors claim that 'the stochastic gradient noise is compensated for by adding less noise to the momentum update.' How is this done? Can authors please explain it a bit more.

3.  For the experiments, the authors introduced ResNet++ architecture which is based on ResNet architecture but is novel in its design with the use of SELU and removal of BatchNorm and with different initialization schemes. The design idea is well explained by the authors. The experiments were performed on CIFAR10 and ImageNet dataset to show the large scale scaling of the approach. A comparison with SGNHT is provided in the paper but a fair comparison with other approaches like SGHMC [3], PSGLD [4] and MCMC [1] is missing which might have proved the effectiveness of the approach when compared to other methods. Any comments on why the experiments were restricted to the ones mentioned in the paper?

4. Lastly, the ResNet++ network is trained on ImageNet and CIFAR10 datasets. There is no clear mention of the time duration it took for training and evaluating the network. The authors mentioned ‘BatchNorm did result in an overhead of roughly 20% compared to the ResNet++ model’, which is not clear. I would urge the authors to explain the above line and provide an estimate of the training time and testing time.

A closing remark: The mention of the acknowledgement section in a double-blind review is not advisable and in future please refrain from doing so.

References:
[1] https://openreview.net/pdf?id=HkeoOo09YX#page=11&zoom=100,0,754
[2] http://people.ee.duke.edu/~lcarin/sgnht-4.pdf
[3] https://arxiv.org/pdf/1402.4102.pdf
[4] https://arxiv.org/abs/1512.07666







**Experience Assessment:**

I have published one or two papers in this area.

**Review Assessment: Checking Correctness Of Derivations And Theory:**

I assessed the sensibility of the derivations and theory.

**Review Assessment: Checking Correctness Of Experiments:**

I carefully checked the experiments.

**Review Assessment: Thoroughness In Paper Reading:**

I read the paper at least twice and used my best judgement in assessing the paper.

---

> ### Author Response · Authors · 2019-11-12
> **Response to Official Blind Review #2**
>
> We thank the reviewer for taking the time to write and elaborate review with detailed questions. We would like to provide answers to the questions raised:
>
> 1) ATMC falls within the framework of SG-MCMC methods and its theoretical soundness relies on the general framework (Eq. 1) to define unbiased SG-MCMC methods.
> In comparison with the meta-learning approach our work scales to much larger models resulting in far better predictive accuracy (78.12% for meta-learning vs. 93.9% for ATMC on Cifar10). However, both algorithms introduce mostly orthogonal changes to the original SGHMC algorithm. A meta-learning approach could be used on top of ATMC to learn the thermostat dynamics as well as the interaction between momentum and parameter variables. However, more research is required to show that meta-learning can scale to large problems like ImageNet.
>
> 2) The amount of noise injected during the momentum update is proportional to $\alpha(\xi)$ which for ATMC is defined to be $\alpha(\xi) = max(0, D_c - \xi)$. Note that if $0 < \xi < D_c$ then $\alpha(\xi) = D_c - \xi$. Thus, the injected noise is the target noise $D_c$ minus the estimate of the gradient noise $\xi$. The fact that $\xi$ is an estimate for the stochastic gradient noise can be seen from the energy function (Eq. 2) which shows that $E[\xi] \propto B$ where $B$ was defined as the covariance of the stochastic gradient noise.
>
> 3) There are indeed many other MCMC methods that one could compare too but for the large scale problems we target some practical issues must be considered. In general, most prior work is limited to small datasets and models and scaling these methods to the problems we are trying to solve is non-trivial. To illustrate, in [1] sampling a model with SGHMC on Cifar10 requires the noise term in SGHMC to be ignored (which reduces SGHMC to SGD+Momentum) for the first 80% of each training cycle. We have tried to limit the experiments to "pure" MCMC methods that always inject the right amount of noise. This way we avoid any doubt about whether these methods can be interpreted as Bayesian Inference algorithms.
>
> SGHMC differs from SGHNT only by the lack of a temperature control variate. Because this sampler is not robust to noise, it requires an explicit estimate of the stochastic gradient noise $B$. Unfortunately, Per-example gradients are poorly supported in modern ML frameworks and naive implementations lead to significant performance regression [3].
>
> Finally, there are preconditioned variants of these algorithms (pSGLD, pSGHMC). Using adaptive preconditioning also leads to bias in practise because the $\Gamma$ correction term becomes intractable [2]. Further note that these methods use the same preconditioner as RMSProp/Adam and that our optimization baselines do not use preconditioning either. Therefore, We consider preconditioned methods to be mostly orthogonal to the improvements we propose and the baseline we compare to.
>
> 4) We agree the the way this was phrased makes the statement ambiguous and we will update this in a revision. Each training step using the BatchNorm model takes 20% longer compared to a single training or sampling step on ResNet++ in wall clock time. The number of training epochs for the respective methods is already mentioned in the paper.
>
> [1]: https://arxiv.org/abs/1902.03932
> [2]: https://arxiv.org/abs/1512.07666
> [3]: https://arxiv.org/pdf/1206.6380.pdf

---

### Official Review · AnonReviewer3 · 2019-10-29
**Official Blind Review #3**

**Rating:** 3

**Review:**

The authors propose the adaptive thermostat Monte Carlo sampler for feedforward neural networks. The proposed approach dynamically adjust the amount of momentum and noisy applied to each model parameter during updates. ResNet++ (ResNet without batchnorm/dropout but adding SELU, fixup and weight normalization) is introduced. Further, the authors claim that the need for hyperparameter setup is reduced provided that early stopping, stochastic regularization and carefully tuned learning rate schedules are not required.

The authors highlight some practical issues with the Nose-hoover thermostat, however, recognize its mathematical soundness. When ATMC is described (temperature stages), a motivation is provided but not justified theoretically.

In (10) and (11), \gamma_1(), \gamma_2(), \eta_t and a are introduced without definition or further explanation.

The authors claim that the need for hyperparameter setup is reduced, however, in the experiments they use a cyclic step size with length n=50 (20 for ImageNet), a Laplace prior with parameter b=5, momentum noise with parameter 0.9, pre-conditioner parameter 0.0003 and c parameter 0.001. The impact of these choices on performance is not described. Further, the number of filters is doubled relative to ResNet-56 without explanation.

The calibration curves in Figure 3 are underwhelming. ATMC is better than SGD but not necessarily well calibrated. Also, note that x and y scales are heavily biased toward 1.

In summary, the proposed approach needs to be described in more detail and the experiments are not very satisfying given the claims made by the authors in the Introduction.

Minor:
- In (1) W is not defined.
- In (1) the dimensionality of D, Q and \Gamma is not defined but their elements are used.
- In (2) m is only defined after (3), in fact, only called by its name, pre-conditioner, in Algorithm 1.
- In Section 2.3 there is a reference to the step size, though not introduced until discretization later in Section 3.
- In (7) \beta() is a function of p, but not in other instances, e.g., (4), (10) and (11).
- Move Algorithm 1 closer to definition.
- In (13), d is not defined.

**Experience Assessment:**

I have published one or two papers in this area.

**Review Assessment: Checking Correctness Of Derivations And Theory:**

I carefully checked the derivations and theory.

**Review Assessment: Checking Correctness Of Experiments:**

I carefully checked the experiments.

**Review Assessment: Thoroughness In Paper Reading:**

I read the paper thoroughly.

---

> ### Author Response · Authors · 2019-11-12
> **Response to Official Blind Review #3**
>
> We thank the reviewer for the thorough review and the useful feedback. We hope to clarify to problems raised by the reviewer:
>
> The theoretical justification for this method is provided by the general framework for constructing SDEs with a target equilibrium distribution introduced in (1). We introduce a generalized thermostat that controls the noise injection using the function $\alpha(\xi)$. The Nose-Hoover thermostat is simply a special case where $\alpha(\xi)$ is a constant. One theoretical justification for why these control variates are useful is that the that they compensate for noisy gradient, see (4). The adaptive thermostat is theoretically sound because it preserves the equilibrium distribution due to (1). Additionally, it improves convergence as is demonstrated empirically and motivated by inspecting the behavior of the dynamics in contrast to the Nose-Hoover thermostat.
>
> Please note that there are many implicit hyper parameters already incorporated in the Cifar and ImageNet baselines we use. For example, the learning rate schedule with the following hyper parameters: a base learning rate; 3 boundary epochs at which the learning rate is reduced by a certain factor; and an early-stopping epoch. All these hyper parameters require careful tuning to avoid over and under fitting issues. In contrast, the sampling method uses an initial learning rate and a cyclic length. Note that making the cycle length longer or the learning rate smaller will only lead to slower convergence but given enough compute will not lead to weaker results or overfitting. The training also no longer requires early stopping and the number of training epochs can instead be chosen as a compute vs. accuracy tradeoff. Other hyper parameters for the optimization baseline include BatchNorm parameters like moving average decay, epsilon, and weight decay. The sampler comes with more hyper parameters (eps, m, c, D) compared to SGD (eps, friction). However, m and c have a strong interpretation as the average and maximum magnitude of parameter updates.
> The doubling of filters was done for all methods compared and should therefore not affect the comparison. The aim of this paper is too scale up Bayesian methods and wider ResNet architecture have been shown previously to improve results [1].
>
> We plot calibration on a log scale as opposed to the typically used linear scale. We believe that this will show more faithfully how much the model probabilities can be trusted in cases where extremely high confidence is required. Do note that for high confidence predictions, even small deviations between frequency and observations lead to large differences on the logarithmic axis. Due to the predictive power of these models the distribution of predictive uncertainty (1 - confidence) is roughly log-uniform. Note that in the domain [0., 0.9] the calibration is almost perfect and the domain [0.99, 1.) would  be nearly imperceptible on the commonly used linear calibration plots.
>
> [1]: https://arxiv.org/abs/1605.07146

---

### Official Review · AnonReviewer4 · 2019-10-30
**Official Blind Review #4**

**Rating:** 6

**Review:**

This paper proposes a novel MCMC algorithm (ATMC) that estimates and samples from the posterior distribution of neural network weights. The motivation for this approach is that applying Bayesian inference to deep learning should lead to less overfitting and better uncertainty-calibrated models. Unlike previous work, the proposed method scales to large models (ResNet) and data sets (ImageNet).

In addition to the main contribution, this work improves an existing numerical integrator necessary for the ATMC sampler. Moreover, in order to apply this method to ResNet, the authors use a modified version of ResNet without stochastic regularization (batch normalization and dropout).

Empirical results show that the proposed method outperforms baselines on accuracy, log-likelihood, and uncertainty calibration. The single-sample posterior already yields decent results, beating a batchnorm-free version of ResNet.

Overall, the paper is well-written, fluently readable, and provides a clear presentation of motivations and methods. Experiments, although not very extensive, are described in sufficient detail and corroborate the claims. Related previous work is cited throughout the paper, although there is no explicit section for it.


Weaknesses:

While the authors claim that the need for hyperparameter tuning is reduced, they use a cyclic step size with parameter n, a Laplace prior with parameter b, a momentum noise with parameter 0.9, and dataset-specific parameters h0, m, and c. This weakens (or even contradicts) the claim, and raises the question of how much the choice of these hyperparameters affects performance.

On ImageNet, ATMC doesn't seem to be well calibrated, and the authors do not discuss the fact that the calibration is basically on par with SGNHT.


Minor:
- page 4: practise -> practice
- page 7: extra space after "compatible with MCMC methods"
- page 7: problem specific -> problem-specific

**Experience Assessment:**

I do not know much about this area.

**Review Assessment: Checking Correctness Of Derivations And Theory:**

I did not assess the derivations or theory.

**Review Assessment: Checking Correctness Of Experiments:**

I assessed the sensibility of the experiments.

**Review Assessment: Thoroughness In Paper Reading:**

I read the paper at least twice and used my best judgement in assessing the paper.

---

> ### Author Response · Authors · 2019-11-12
> **Response to Official Blind Review #4**
>
> We thank the reviewer for the accurate summary of the paper and the detailed feedback provided. We would like to elaborate on the issues concerning hyper parameters and calibration raised by the reviewer.
>
> Please note that there are many implicit hyper parameters already incorporated in the Cifar and ImageNet baselines we use. For example, the learning rate schedule with the following hyper parameters: a base learning rate; 3 boundary epochs at which the learning rate is reduced by a factor 10; and an early-stopping epoch. All these hyper parameters require careful tuning to avoid over and under fitting issues. In contrast, the sampling method uses an initial learning rate and a cyclic length. Note that making the cycle length longer or the learning rate smaller will only lead to slower convergence but given enough compute will not lead to weaker results or overfitting. The training also no longer requires early stopping and the number of training epochs can instead be chosen as a compute vs. accuracy tradeoff. Other hyper parameters for the optimization baseline include BatchNorm parameters like moving average decay, epsilon, and weight decay. The sampler comes with more hyper parameters (eps, m, c, D) compared to SGD (eps, friction). However, m and c have a strong interpretation as the average and maximum magnitude of parameter updates.
>
> We plot calibration on a log scale as opposed to the typically used linear scale. We believe that this will show more faithfully how much the model probabilities can be trusted in cases where extremely high confidence is required. Do note that for high confidence predictions, even small deviations between frequency and observations lead to large differences on the logarithmic axis. We agree that SGNHT calibration is indeed competitive with ATMC. Do however note that our results on SGHNT include the improvements on the ResNet model and numerical integration and are, to the best of our knowledge, significantly better than previously reported results. Also note that calibration is an easier task when the predictive accuracy is lower (consider the extreme case: a model predicting just marginals is perfectly calibrated).

---

### Decision · Program_Chairs · 2019-12-19

**Decision:**

Reject

**Comment:**

This paper proposes a variant of Hamiltonian Monte Carlo for Bayesian inference in deep learning.

Although the reviewers acknowledge the ambition, scope and novelty of the paper they still have a number of reservations regarding experimental results and claims (regarding need for hyperparameter tuning). The overall score consequently falls below acceptance.

Rejection is recommended. These reservations made by the referees should definitely be addressable before next conference deadline so looking forward to see the paper published asap.